# Confidence sequences for
# sampling without replacement

**Ian Waudby-Smith**[1] **and Aaditya Ramdas**[12]

Departments of Statistics[1] and Machine Learning[2]
Carnegie Mellon University
{ianws, aramdas}@cmu.edu

## Abstract

Many practical tasks involve sampling sequentially without replacement (WoR) from a finite population of size $N$, in an attempt to estimate some parameter $\theta^\star$. Accurately quantifying uncertainty throughout this process is a nontrivial task, but is necessary because it often determines when we stop collecting samples and confidently report a result. We present a suite of tools for designing *confidence sequences* (CS) for $\theta^\star$. A CS is a sequence of confidence sets $(C_n)_{n=1}^N$, that shrink in size, and all contain $\theta^\star$ simultaneously with high probability. We present a generic approach to constructing a frequentist CS using Bayesian tools, based on the fact that the ratio of a prior to the posterior at the ground truth is a martingale. We then present Hoeffding- and empirical-Bernstein-type time-uniform CSs and fixed-time confidence intervals for sampling WoR, which improve on previous bounds in the literature and explicitly quantify the benefit of WoR sampling.

## 1 Introduction

When data are collected sequentially rather than in a single batch with a fixed sample size, many classical statistical tools cannot naively be used to calculate uncertainty as more data become available. Doing so can quickly lead to overconfident and incorrect results (informally, "peeking, $p$-hacking"). For these kinds of situations, the analyst would ideally have access to procedures that allow them to:

(a) Efficiently calculate tight confidence intervals whenever new data become available;

(b) Track the intervals, and use them to decide whether to continue sampling, or when to stop;

(c) Have valid confidence intervals (or $p$-values) at arbitrary data-dependent stopping times.

The desire for methods satisfying (a), (b), and (c) led to the development of *confidence sequences* (CS) — sequences of confidence sets which are uniformly valid over a given time horizon $\mathcal{T}$. Formally, a sequence of sets $\{C_t\}_{t\in\mathcal{T}}$ is a $(1-\alpha)$-CS for some parameter $\theta^\star$ if

$$\Pr(\forall t \in \mathcal{T},\ \theta^\star \in C_t) \geqslant 1 - \alpha \quad \equiv \quad \Pr(\exists t \in \mathcal{T} : \theta^\star \notin C_t) \leqslant \alpha. \tag{1.1}$$

Critically, (1.1) holds iff $\Pr(\theta^\star \notin C_\tau) \leqslant \alpha$ for arbitrary stopping times $\tau$ [1], yielding property (c). The foundations of CSs were laid by Robbins, Darling, Siegmund & Lai [2, 3, 4, 5]. The multi-armed bandit literature sometimes calls them 'anytime' confidence intervals [6, 7]. CSs have recently been developed for a variety of nonparametric problems [1, 8, 9].

This paper derives closed-form CSs when samples are drawn without replacement (WoR) from a finite population. The technical underpinnings are novel (super)martingales for both categorical (Section 2) and continuous (Section 3) observations. In the latter setting, our results unify and improve on the time-uniform with-replacement extensions of Hoeffding's [10] and empirical Bernstein's inequalities by Maurer and Pontil [11] that have been derived recently [12, 1], with several related inequalities for sampling WoR by Serfling [13] and extensions by Bardenet and Maillard [14] and Greene and Wellner [15].

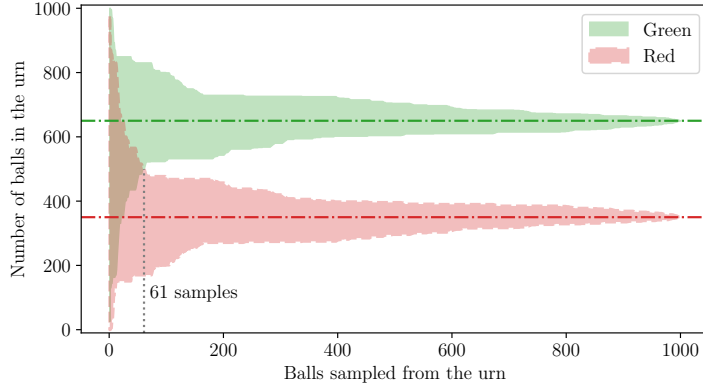

Figure 1: 95% CS for the number of green and red balls in an urn by sampling WoR[2]. Notice that the true totals (650 green, 350 red) are captured by the CSs uniformly over time from the initial sample until all 1000 balls are observed. After sampling 61 balls in this example, the CSs cease to overlap, and we can conclude with 95% confidence that there are more green than red balls in the urn.

**Outline.** In Section 2, we use Bayesian ideas to obtain frequentist CSs for categorical observations. In Section 3, we construct CSs for the mean of a finite set of bounded real numbers. We discuss implications for testing in Section 4. Some prototypical applications are described in Appendix A. The other appendices contain proofs, choices of tuning parameters, and computational considerations.

### 1.1 Notation, supermartingales and the model for sampling WoR

Everywhere in this paper, the $N$ objects in the finite population $\{x_1, \ldots, x_N\}$ are fixed and non-random. In the discrete setting (Section 2) with $K \geq 2$ categories $\{c_k\}_{k=1}^K$, we have $x_i \in \{c_1, c_2, \ldots, c_K\}$. In the continuous setting (Section 3), $x_i \in [\ell, u]$ for some known bounds $\ell < u$. What is random is only the order of observation; the model for sampling uniformly at random WoR posits that

$$X_t \mid \{X_1, \ldots, X_{t-1}\} \sim \text{Uniform}(\{x_1, \ldots, x_N\} \backslash \{X_1, \ldots, X_{t-1}\}). \qquad (1.2)$$

All probabilities in this paper are to be understood as solely arising from observing fixed entities in a random order, with no distributional assumptions being made on the finite population. It is worth remarking on the power of this randomization—as demonstrated in our experiments, one can estimate the average of a deterministic set of numbers to high accuracy without observing a large fraction of the set.

The results in this paper draw from the theory of *supermartingales*. While they can be defined in more generality, we provide a definition of supermartingales which will suffice for the theorems that follow.

A filtration is an increasing sequence of sigma fields. For the entirety of this paper, we consider the 'canonical' filtration $(\mathcal{F}_t)_{t=0}^N$ defined by $\mathcal{F}_t := \sigma(X_1, \ldots, X_t)$, with $\mathcal{F}_0$ is the empty or trivial sigma field. For any fixed $N \in \mathbb{N}$, a stochastic process $(M_t)_{t=0}^N$ is said to be a *supermartingale* with respect to $(\mathcal{F}_t)_{t=0}^N$ if for all $t \in \{0, 1, \ldots, N-1\}$, $M_t$ is measurable with respect to $\mathcal{F}_t$ (informally, $M_t$ is a function of $X_1, \ldots, X_t$), and

$$\mathbb{E}(M_{t+1} \mid \mathcal{F}_t) \leq M_t.$$

If the above inequality is replaced by an equality for all $t$, then $(M_t)_{t=0}^N$ is said to be a *martingale*. For succinctness, we use the notation $a_1^t := \{a_1, \ldots, a_t\}$ and $[a] := \{1, \ldots, a\}$. Using this terminology, one can rewrite model (1.2) as positing that $X_t \mid \mathcal{F}_{t-1} \sim \text{Uniform}(x_1^N \backslash X_1^{t-1})$.

## 2 Discrete categorical setting

When observations are of this discrete form, the variables can be rewritten in such a way that they follow a hypergeometric distribution. In such a setting, the following "prior-posterior-ratio martingale" can be used to obtain CSs for parameters of the hypergeometric distribution which shrink to a single point after all data have been observed.

## 2.1 The prior-posterior-ratio (PPR) martingale

While the PPR martingale will be particularly useful for obtaining CSs when sampling discrete categorical random variables WoR from a finite population, it may be employed whenever one is able to compute a posterior distribution, and is certainly *not limited to this paper's setting*. Moreover, this posterior distribution need not be computed in closed form, and computational techniques such as Markov Chain Monte Carlo may be employed when a conjugate prior is not available or desirable.

To avoid confusion, we emphasize that while we make use of terminology from Bayesian inference such as posteriors and conjugate priors, all of the probability statements with regards to CSs should be read in the frequentist sense, and are not interpreted as sequences of credible intervals.

Consider any family of distributions $\{F_\theta\}_{\theta \in \Theta}$ with density $f_\theta$ with respect to some underlying common measure (such as Lebesgue for continuous cases, counting measure for discrete cases). Let $\theta^\star \in \Theta$ be a fixed parameter and let $\mathcal{T} = [N]$ where $N \in \mathbb{N} \cup \{\infty\}$. Suppose that $X_1 \sim f_{\theta^\star}(x)$ and

$$X_{t+1} \sim f_{\theta^\star}(x \mid X_1^t) \quad \text{for all } t \in \mathcal{T}.$$

Let $\pi_0(\theta)$ be a prior distribution on $\Theta$, with posterior given by

$$\pi_t(\theta) = \frac{\pi_0(\theta) f_\theta(X_1^t)}{\int_{\eta \in \Theta} \pi_0(\eta) f_\eta(X_1^t) d\eta}.$$

To prepare for the result that follows, define the *prior-posterior ratio (PPR)* evaluated at $\theta \in \Theta$ as

$$R_t(\theta) := \frac{\pi_0(\theta)}{\pi_t(\theta)}.$$

**Proposition 2.1** (Prior-posterior-ratio martingale). *For any prior $\pi_0$ on $\Theta$ that assigns nonzero mass everywhere, the sequence of prior-posterior ratios evaluated at the true $\theta^\star$, that is $(R_t(\theta^\star))_{t=0}^N$, is a nonnegative martingale with respect to $(\mathcal{F}_t)_{t=0}^N$. Further, the sequence of sets*

$$C_t := \{\theta \in \Theta : R_t(\theta) < 1/\alpha\}$$

*forms a $(1-\alpha)$-CS for $\theta^\star$, meaning that $\Pr(\exists t \in \mathcal{T} : \theta^\star \notin C_t) \leqslant \alpha$.*

The proof is given in Appendix B.1.

Going forward, we adopt the label *working* before 'prior' and 'posterior' and encase them in 'quotes' to emphasize that they constitute part of a Bayesian 'working model', to contrast it against an assumed Bayesian model; the latter would be inappropriate given the discussion in Section 1.1. Next, we apply this result to the hypergeometric distribution. We will later examine the practical role of this working prior.

## 2.2 CSs for binary settings using the hypergeometric distribution

Recall that a random variable $X$ has a hypergeometric distribution with parameters $(N, N^+, n)$ if it represents the number of "successes" in $n$ random samples WoR from a population of size $N$ in which there are $N^+$ such successes, and each observation is either a success or failure (1 or 0). The probability of a particular number of successes $x \in \{0, 1, \ldots, \min(N^+, n)\}$ is

$$\Pr(X = x) = \binom{N^+}{x}\binom{N - N^+}{n - x}/\binom{N}{n}.$$

For notational simplicity, we consider the case when $n = 1$, that is we make one observation at a time, but this is not a necessary restriction. In fact, one would obtain the same CS at time ten if we repeatedly make one observation ten times, or make ten observations in one go. For a moment, let us view this problem from the Bayesian perspective, treating the fixed parameter $N^+$ as a random parameter, which we call $\widetilde{N}^+$ to avoid confusion. We choose a beta-binomial 'working prior' on $\widetilde{N}^+$ as it is conjugate to the hypergeometric distribution up to a shift in $\widetilde{N}^+$ [16]. Concretely, suppose

$$X_t \mid (\widetilde{N}^+, X_1, \ldots, X_{t-1}) \sim \text{HyperGeo}\left(N - (t-1), \widetilde{N}^+ - \sum_{i=1}^{t-1} X_i, 1\right),$$

$$\widetilde{N}^+ \sim \text{BetaBin}(N, a, b),$$

for some $a, b > 0$. Then for any $t \in [N]$, the 'working posterior' for $\widetilde{N}^+$ is given by

$$\widetilde{N}^+ - \sum_{i=1}^t X_i \mid X_1^t \sim \text{BetaBin}\left(N - t, a + \sum_{i=1}^t X_i, b + t - \sum_{i=1}^t X_i\right).$$

Now that we have 'prior' and 'posterior' distributions for $\widetilde{N}^+$, an application of the prior-posterior martingale (Proposition 2.1) yields a CS for the true $N^+$, summarized in the following theorem.

**Theorem 2.1** (CS for binary observations). *Suppose $x_1^N \in \{0,1\}^N$ is a nonrandom set with the number of successes $\sum_{i=1}^N x_i \equiv N^+$ fixed and unknown. Under observation model (1.2), we have*

$$X_t \mid X_1^{t-1} \sim HyperGeo\left(N - (t-1), N^+ - \sum_{i=1}^{t-1} X_i, 1\right).$$

*For any beta-binomial 'prior' $\pi_0$ for $N^+$ with parameters $a, b > 0$ and induced 'posterior' $\pi_t$,*

$$C_t := \left\{ n^+ \in [N] : \frac{\pi_0(n^+)}{\pi_t(n^+)} < \frac{1}{\alpha} \right\}$$

*is a $(1-\alpha)$-CS for $N^+$. Further, the running intersection, $(\bigcap_{s \leq t} C_t)_{t \in [N]}$ is also a valid $(1-\alpha)$-CS.*

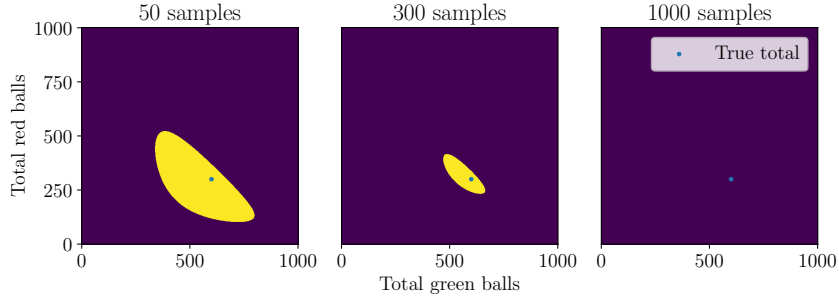

Figure 2: Consider sampling balls from an urn WoR with three distinct colors (red, green, and purple). In this example, the urn contains 1000 balls with 300 red, 600 green, and 100 purple. We only require a two-dimensional confidence sequence (yellow region) to capture uncertainty about all three totals. After around 300 balls have been sampled, we are quite confident that the urn is made up mostly of green; after 1000 samples, we know the totals for each color with certainty.

The proof of Theorem 2.1 is a direct application of Proposition 2.1. Note that for any 'prior', the 'posterior' at time $t = N$ is $\pi_N(n^+) = \mathbb{1}(n^+ = N^+)$, so $C_t$ shrinks to a point, containing only $N^+$. For $K > 2$ categories, Theorem 2.1 can be extended to use a multivariate hypergeometric with a Dirichlet-multinomial prior to yield higher-dimensional CSs, but we leave the (notationally heavy) derivation to Appendix C. See Figure 2 to get a sense of what these CSs can look like when $K = 3$.

### 2.3 Role of the 'prior' in the prior-posterior CS

The prior-posterior CSs discussed thus far have valid (frequentist) coverage for any 'prior' on $N^+$, and in particular are valid for a beta-binomial 'prior' with any data-independent choices of $a, b > 0$. Importantly, the corresponding CS always shrinks to zero width. How, then, should the user pick $(a, b)$? Figure 3 provides some visual intuition.

These are our takeaway messages: (a) if the 'prior' is very accurate (coincidentally peaked at the truth), the resulting CS is narrowest, (b) even if the 'prior' is horribly inaccurate (placing almost no mass at the truth), the resulting CS is well-behaved and robust, albeit wider, (c) if we do not actually have any idea what the underlying truth might be, we suggest using a uniform 'prior' to safely balance the two extremes. However, a more risky 'prior' pays a relatively low statistical price.

## 3 Bounded real-valued setting

Suppose now that observations are real-valued and bounded as in Examples C and D of Appendix A. Here we introduce Hoeffding- and empirical Bernstein-type inequalities for sampling WoR.

### 3.1 Hoeffding-type bounds

Recalling Section 1.1, we deal with a fixed batch $x_1^N$ of bounded real numbers $x_i \in [\ell, u]$ with mean $\mu := \frac{1}{N} \sum_{i=1}^N x_i$. Our CS for $\mu$ will utilize a novel WoR mean estimator,

$$\widehat{\mu}_t := \frac{\sum_{i=1}^t X_i + \sum_{i=1}^t \frac{1}{N-i+1} \sum_{j=1}^{i-1} X_j}{t + \sum_{i=1}^t \frac{i-1}{N-i+1}}. \tag{3.1}$$

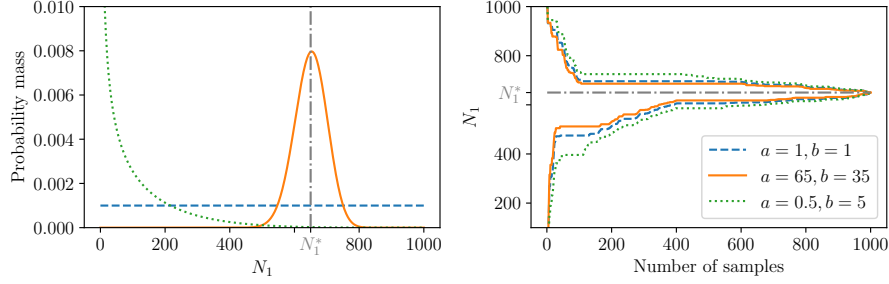

Figure 3: Beta-binomial probability mass function as a 'prior' on $N_1^\star$ with different choices of $(a, b)$, and the resulting PPR CS for the parameter $N_1^\star$ of a hypergeometric distribution when $(N_1^\star, N_2^\star) = (650, 350)$.

More generally, if $\lambda_1, \ldots, \lambda_N$ is a predictable sequence (meaning $\lambda_t$ is $\mathcal{F}_{t-1}$-measurable for $t \in \{1, \ldots, N\}$), then we may define the weighted WoR mean estimator,

$$\widehat{\mu}_t(\lambda_1^t) := \frac{\sum_{i=1}^{t} \lambda_i (X_i + \frac{1}{N-i+1} \sum_{j=1}^{i-1} X_j)}{\sum_{i=1}^{t} \lambda_i (1 + \frac{i-1}{N-i+1})}, \tag{3.2}$$

where it should be noted that if $\lambda_1 = \cdots = \lambda_N$ then $\widehat{\mu}_t(\lambda_1^t)$ recovers $\widehat{\mu}_t$. Past WoR works [13, 14, 15] base their bounds on the sample average $\sum_i X_i/t$. Both $\widehat{\mu}_t$ and the sample average are conditionally biased and unconditionally unbiased (see Appendix B.2 for more details). As frequently encountered in Hoeffding-style inequalities for bounded random variables [10], define

$$\psi_H(\lambda) := \frac{\lambda^2 (u - \ell)^2}{8}. \tag{3.3}$$

Setting $M_0^H := 1$, we introduce a new exponential Hoeffding-type process for a predictable sequence $\lambda_1^N$,

$$M_t^H := \exp\left\{ \sum_{i=1}^{t} \left[ \lambda_i \left( X_i - \mu + \frac{1}{N-i+1} \sum_{j=1}^{i-1} (X_j - \mu) \right) - \psi_H(\lambda_i) \right] \right\}. \tag{3.4}$$

**Theorem 3.1** (A time-uniform Hoeffding-type CS for sampling WoR). *Under the observation model and filtration $(\mathcal{F}_t)_{t=0}^{N}$ of Section 1.1, and for any predictable sequence $\lambda_1^N$, the process $(M_t^H)_{t=0}^{N}$ is a nonnegative supermartingale, and thus,*

$$\Pr\left( \exists t \in [N] : \mu - \widehat{\mu}_t(\lambda_1^t) \geqslant \frac{\sum_{i=1}^{t} \psi_H(\lambda_i) + \log(1/\alpha)}{\sum_{i=1}^{t} \lambda_i \left(1 + \frac{i-1}{N-i+1}\right)} \right) \leqslant \alpha.$$

*Consequently,*

$$C_t^H := \widehat{\mu}_t(\lambda_1^t) \pm \frac{\sum_{i=1}^{t} \psi_H(\lambda_i) + \log(2/\alpha)}{\sum_{i=1}^{t} \lambda_i \left(1 + \frac{i-1}{N-i+1}\right)} \quad \text{forms a } (1-\alpha)\text{-CS for } \mu.$$

The proof in Appendix B.2 combines ideas from the with-replacement, *time-uniform* extension of Hoeffding's inequality of Howard et al. [1, 12] with the fixed-time, *without-replacement* extension of Hoeffding's by Bardenet & Maillard [14], to yield a bound that improves on both. When $\lambda := \lambda_1 = \cdots = \lambda_N$ is a constant, the term

$$A_t := \sum_{i=1}^{t} \frac{i-1}{N-i+1} \tag{3.5}$$

captures the 'advantage' over the classical Hoeffding's inequality; we discuss this term more soon.

In order to use the aforementioned CS, one needs to choose a predictable $\lambda$-sequence. First, consider the simpler case of a fixed real-valued $\lambda := \lambda_1 = \cdots \lambda_N$ as this will aid our intuition in choosing a more complex $\lambda$-sequence. In this case, $\lambda$ corresponds to a time $t_0 \in [N]$ for which the CS is

tightest. If the user wishes to optimize the width of the CS for time $t_0$, then the corresponding $\lambda$ to be used is given by

$$\lambda := \sqrt{\frac{8\log(2/\alpha)}{t_0(u-\ell)^2}}. \tag{3.6}$$

Alternatively, if the user does not wish to commit to a single time $t_0$, they can choose a $\lambda$-sequence akin to (3.6) but which spreads its width optimization over time. For example, one can use the sequence for $t \in \{1, \dots, N\}$,

$$\lambda_t := \sqrt{\frac{8\log(2/\alpha)}{t\log(t+1)(u-\ell)^2}} \wedge \frac{1}{u-\ell}, \tag{3.7}$$

where the minimum was taken to prevent the CS width from being dominated by early terms. Note however that any predictable $\lambda$-sequence yields a valid CS (see Appendix E for more examples).

Optimizing a real-valued $\lambda = \lambda_1 = \cdots = \lambda_N$ for a particular time is in fact the typical strategy used to obtain the tightest fixed-time (i.e. non-sequential) Chernoff-based confidence intervals (CIs) such as those based on Hoeffding's inequality [1, 10]. This same strategy can be used with our WoR CSs to obtain tight fixed-time CIs for sampling WoR. Specifically, plugging (3.6) into Theorem 3.1 for a fixed sample size $n \in [N]$, we obtain the following corollary.

**Corollary 3.1** (Hoeffding-type CI for sampling WoR). *For any $n \in [N]$,*

$$\widehat{\mu}_n \pm \frac{\sqrt{\frac{1}{2}(u-\ell)^2\log(2/\alpha)}}{\sqrt{n} + A_n/\sqrt{n}} \quad \textit{forms a } (1-\alpha) \textit{ CI for } \mu. \tag{3.8}$$

Notice that the classical Hoeffding confidence interval is recovered exactly, including constants, by dropping the $A_n$ term and using the usual sample mean estimator instead of $\widehat{\mu}_t$. To get a sense of how large the advantage is, note that

$$\text{for small } n \ll N, \quad A_n \asymp \sum_{i=1}^{n-1} i/N \asymp n^2/N,$$

$$\text{for large } n \approx N, \quad A_n \asymp A_N = \sum_{i=1}^{N-1} \frac{i}{N-i} = \sum_{j=1}^{N-1} \frac{N-j}{j} \asymp N\log N - (N-1).$$

Thus, the advantage is negligible for $n = O(\sqrt{N})$, while it is substantial for $n = O(N)$, but it is clear that the CI of (3.8) is strictly tighter than Hoeffding's inequality for any $n$.

## 3.2 Empirical Bernstein-type bounds

Hoeffding-type bounds like the one in Theorem 3.1 only make use of the fact that observations are bounded, and they can be loose if only some observations are near the boundary of $[\ell, u]$ while the rest are concentrated near the middle of the interval. More formally, the CS of Theorem 3.1 has the same width whether the underlying population $x_1^N$ has large or small variance $\sum_{i=1}^N (x_i - \mu)^2$— thus, they are tightest when the $x_i$s equal $\ell$ or $u$, and they are loosest when $x_i \approx (\ell + u)/2$ for all $i$. As an alternative that adaptively takes a variance-like term into account [11, 17], we introduce a sequential, WoR, empirical Bernstein CS. As is typical in empirical Bernstein bounds [1], we use a different 'subexponential'-type function,

$$\psi_E(\lambda) := (-\log(1-c\lambda) - c\lambda)/4 \quad \text{for any } \lambda \in [0, 1/c)$$

where $c := u - \ell$. $\psi_E$ seems quite different from $\psi_H$, but Taylor expanding log yields $\psi_E(\lambda) \approx c^2\lambda^2/8$. Indeed,

$$\lim_{\lambda \to 0} \psi_E(\lambda)/\psi_H(\lambda) = 1. \tag{3.9}$$

Note that one typically picks small $\lambda$, e.g.: set $t_0 = N/2, \ell = -1, u = 1$ in (3.6) to get $\lambda_1 \propto 1/\sqrt{N}$. In what follows, we derive a time-uniform empirical-Bernstein inequality for sampling WoR. Similar to Theorem 3.1, underlying the bound is an exponential supermartingale. Set $M_0^E = 1$, and recall that $c = u - \ell$ to define a novel exponential process for any $[0, 1/c)$-valued predictable sequence $\lambda_1, \dots \lambda_N$:

$$M_t^E := \exp\left\{\sum_{i=1}^{t}\left[\lambda_i\left(X_i - \mu + \frac{1}{N-i+1}\sum_{j=1}^{i-1}(X_j - \mu)\right) - \left(\frac{c}{2}\right)^{-2}(X_i - \widehat{\mu}_{i-1})^2\psi_E(\lambda_i)\right]\right\}. \tag{3.10}$$

**Theorem 3.2** (A time-uniform empirical Bernstein-type CS for sampling WoR). *Under the observation model and filtration $(\mathcal{F}_t)_{t=0}^N$ of Section 1.1, and for any $[0, 1/c)$-valued predictable sequence $\lambda_1^N$, the process $(M_t^E)_{t=0}^N$ is a nonnegative supermartingale, and thus,*

$$\Pr\left(\exists t \in [N] : \mu - \widehat{\mu}_t(\lambda_1^t) \geqslant \frac{\sum_{i=1}^t (c/2)^{-2} (X_i - \widehat{\mu}_{i-1})^2 \psi_E(\lambda_i) + \log(1/\alpha)}{\sum_{i=1}^t \lambda_i \left(1 + \frac{i-1}{N-i+1}\right)}\right) \leqslant \alpha.$$

*Consequently,*

$$C_t^E := \widehat{\mu}_t(\lambda_1^t) \pm \frac{\sum_{i=1}^t (c/2)^{-2} (X_i - \widehat{\mu}_{i-1})^2 \psi_E(\lambda_i) + \log(2/\alpha)}{\sum_{i=1}^t \lambda_i \left(1 + \frac{i-1}{N-i+1}\right)} \quad \text{forms a } (1-\alpha)\text{-CS for } \mu.$$

The proof in Appendix B.3 involves modifying the proof of Theorem 4 in Howard et al. [1] to use our WoR versions of $\widehat{\mu}_t$ and to include predictable values of $\lambda_t$.

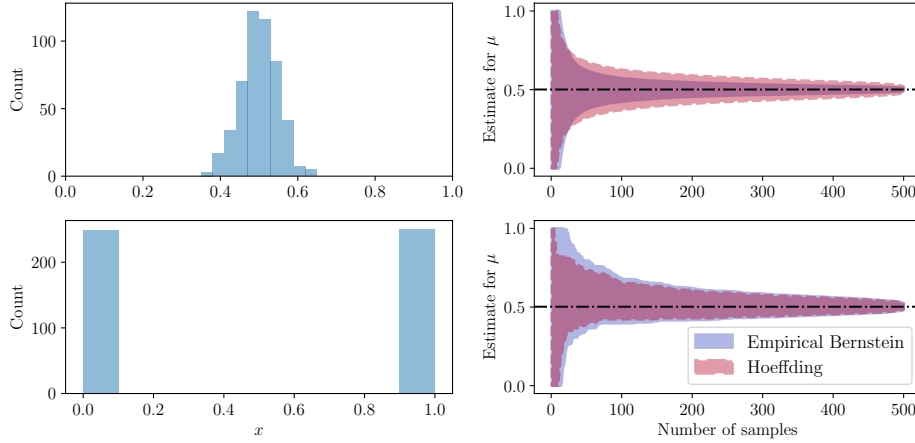

Figure 4: Left-most plots show the histogram of the underlying set of numbers $x_1^N \in [0, 1]^N$, while right-most plots compare empirical Bernstein- and Hoeffding-type CSs for $\mu$. Specifically, the Hoeffding and empirical Bernstein CSs use the $\lambda$-sequences in (3.7) and (3.13), respectively. As expected, in low-variance settings (top), $C_t^E$ is superior, but in a high-variance setting (bottom), $C_t^H$ has a slight edge.

As before one must choose a $\lambda$-sequence to use $C_t^E$. We will again consider the case of a real-valued $\lambda := \lambda_1 = \cdots \lambda_N$ to help guide our intuition on choosing a more complex $\lambda$-sequence. Unlike earlier, we cannot optimize the width of $C_t^E$ in closed-form since $\psi_E$ is less analytically tractable. Once more, fact (3.9) comes to our rescue: substituting $\psi_H$ for $\psi_E$ and optimizing the width yields an expression like (3.6):

$$\lambda^\star := \sqrt{\frac{2 \log(2/\alpha)}{\widehat{V}_t}}, \tag{3.11}$$

where $\widehat{V}_t := \sum_{i=1}^t (X_i - \widehat{\mu}_{i-1})^2$ is a variance process. However, we cannot use this choice of $\lambda^\star$ since it depends on $X_1^t$. Instead, we construct a predictable $\lambda$-sequence which mimics $\lambda^\star$ and adapts to the underlying variance as samples are collected. To heuristically optimize the CS for a particular time $t_0$, take an estimate $\widehat{\sigma}_{t-1}^2$ of the variance which only depends on $X_1^{t-1}$, and set

$$\lambda_t := \sqrt{\frac{2 \log(2/\alpha)}{\widehat{\sigma}_{t-1}^2 t_0}} \wedge \frac{1}{2c}. \tag{3.12}$$

Alternatively, to spread the CS width optimization over time as in (3.7), one can use the $\lambda$-sequence,

$$\lambda_t := \sqrt{\frac{2 \log(2/\alpha)}{\widehat{\sigma}_{t-1}^2 t \log(t+1)}} \wedge \frac{1}{2c}, \tag{3.13}$$

but again, any predictable sequence will suffice.

Similarly to the Hoeffding-type CS, we may instantiate the empirical Bernstein-type CS at a particular time to obtain tight CIs for sampling WoR. However, ensuring that the resulting fixed-time CI is valid when using a data-dependent $\lambda$-sequence requires some additional care. Suppose now that $X_1^n$ is a simple random sample WoR from the finite population, $x_1^N \in [\ell, u]^N$. If we randomly permute $X_1, \ldots, X_n$ to obtain the sequence, $\widetilde{X}_1, \ldots, \widetilde{X}_n$, we have recovered the observation model of Section 1.1, and thus Theorem 3.2 applies. We choose a $\lambda$-sequence which sequentially estimates the variance, but heuristically optimizes for the sample size $n$ as in (3.12). For $t \in [n]$, define

$$\widetilde{\lambda}_t := \sqrt{\frac{2\log(2/\alpha)}{n\widetilde{\sigma}_{t-1}^2}} \wedge \frac{1}{2c} \quad \text{where} \quad \widetilde{\sigma}_t^2 := \frac{c^2/4 + \sum_{i=1}^t (\widetilde{X}_i - \widetilde{\mu}_i)^2}{t+1} \quad \text{and} \quad \widetilde{\mu}_t := \frac{1}{t}\sum_{i=1}^t \widetilde{X}_i. \quad (3.14)$$

Here, an extra $c^2/4$ was added to $\widetilde{\sigma}_t^2$ so that it is defined at time 0, but this is simply a heuristic and any other choice of $\widetilde{\sigma}_0^2$ will suffice. The resulting CI can be summarized in the following corollary.

**Corollary 3.2.** *Let $X_1^n$ be a simple random sample WoR from the finite population $x_1^N$ and let $\widetilde{X}_1^n$ be a random permutation of $X_1^n$. Let $\widetilde{\lambda}_t$ be a predictable sequence such as the one in (3.14) for each $t \in [n]$. Then for any $n \in [N]$,*

$$\widehat{\mu}_n(\widetilde{\lambda}_1^n) \pm \frac{\sum_{i=1}^n (c/2)^{-2}(\widetilde{X}_i - \widetilde{\mu}_{i-1})^2 \psi_E(\widetilde{\lambda}_i) + \log(2/\alpha)}{\sum_{i=1}^n \widetilde{\lambda}_i \left(1 + \frac{i-1}{N-i+1}\right)} \text{ forms a } (1-\alpha) \text{ CI for } \mu.$$

The aforementioned CSs and CIs have a strong relationship with corresponding hypothesis tests. In the following section, we discuss how one can use the techniques developed here to sequentially test hypotheses about finite sets of nonrandom numbers.

# 4 Testing hypotheses about finite sets of nonrandom numbers

In classical hypothesis testing, one has access to i.i.d. data from some underlying distribution(s), and one wishes to test some property about them; this includes sequential tests dating back to Wald [18]. However, it is not often appreciated that it is possible to test hypotheses about a finite list of numbers that do not have any distribution attached to them. Recalling the setup of Section 1.1, this is the nonstandard setting we find ourselves in. For instance in the same example as Figure 1, we may wish to test:

$$H_0 : N_1^\star \leqslant 550 \quad \text{(At most 550 of the balls are green).}$$

If we had access to each ball in advance, then we could accept or reject the null without any type-I or type-II error, but this is tedious, and so we sequentially take samples in a random order to test this hypothesis. The main question then is: *how do we calculate a $p$-value $P_t$ that we can track over time, and stop sampling when $P_t \leqslant 0.05$?*

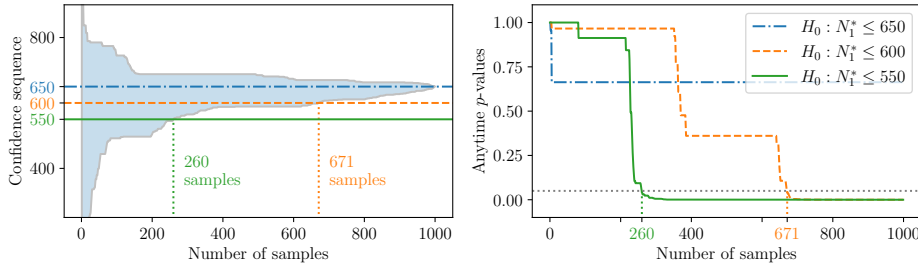

Figure 5: The duality between anytime $p$-values and CSs for three null hypotheses: $H_0 : N_1^\star \leqslant D$ for $D \in \{550, 600, 650\}$. The first null is rejected at a $5\%$ significance level after 260 samples, exactly when the $95\%$ CS stops intersecting the null set $[0, 550]$. However, $H_0 : N_1^\star \leqslant 650$ is never rejected since 650, the ground truth, is contained in the CS at all times from 0 to 1000.

Luckily, we do not need any new tools for this, and our CSs provide a straightforward answer. Though we left it implicit, each confidence sequence $C_t$ is really a function of confidence level $\alpha$. Consider the family $\{C_t(q)\}_{q \in (0,1)}$ indexed by $q$, which we only instantiated at $q = \alpha$. Now, define

$$P_t := \inf\{q : C_t(q) \cap H_0 = \varnothing\}, \quad (4.1)$$

which is the smallest error level $q$ at which $C_t(q)$ just excludes the null set $H_0$. This 'duality' is familiar in non-sequential settings, and in our case it yields an anytime-valid $p$-value [19, 1],

$$\text{Under } H_0, \quad \Pr(\exists t \in [N] : P_t \leqslant \alpha) \leqslant \alpha \text{ for any } \alpha \in [0, 1].$$

In words, if the null hypothesis is true, then $P_t$ will remain above $\alpha$ through the whole process, with probability $\geqslant 1 - \alpha$. To more clearly bring out the duality to CSs, define the stopping time

$$\tau := \inf\{t \in [N] : P_t \leqslant \alpha\}, \text{ and we set } \tau = N \text{ if the inf is not achieved.}$$

Then under the null, $\tau = N$ (we never stop early) with probability $\geqslant 1 - \alpha$. If we do stop early, then $\tau$ is exactly the time at which $C_t(\alpha)$ excluded the null set $H_0$. The manner in which anytime-valid $p$-values and CSs are connected through stopping times is demonstrated in Figure 5.

In summary, our CSs directly yield $p$-values (4.1) for composite null hypotheses. These $p$-values can be tracked, and are valid simultaneously at all times, including at arbitrary stopping times. Aforementioned type-I error probabilities are due to the randomness in the ordering, not in the data. It is worth noting that our (super)martingales $(R_t)$, $(M_t^H)$ and $(M_t^E)$ also immediately yield 'e-values' [20] and hence 'safe tests' [21], meaning that under nulls of the form in Figure 5, they satisfy $\mathbb{E}M_\tau \leqslant 1$ for any stopping time $\tau$.

## 5   Summary

WoR sampling and inference naturally arise in a variety of applications such as finite-population studies and permutation-based statistical methods as outlined in Appendix A. Furthermore, several machine learning tasks involve random samples from finite 'populations', such as sampling (a) points for a stochastic gradient method, (b) covariates in a random order for coordinate descent, (c) columns of a matrix, or (d) edges in a graph.

In order to quantify uncertainty when sequentially sampling WoR from a finite set of objects, this paper developed three new confidence sequences: one in the discrete setting and two in the continuous setting (Hoeffding, empirical-Bernstein). Their construction was enabled by the development of new technical tools—the prior-posterior-ratio martingale, and two exponential supermartingales—which may be of independent interest. We clarified how these can be tuned (role of 'prior' or $\lambda$-sequence), and demonstrated their advantages over naive sampling with replacement. Our CSs can be inverted to yield anytime-valid $p$-values to sequentially test arbitrary composite hypotheses. Importantly, these CSs can be efficiently updated, continuously monitored, and adaptively stopped without violating their uniform validity, thus merging theoretical rigor with practical flexibility.

### Acknowledgements

IW-S thanks Serge Aleshin-Guendel for conversations regarding Bayesian methods. AR thanks Steve Howard for early conversations. AR acknowledges funding from an Adobe Faculty Research Award, and an NSF DMS 1916320 grant.

### Broader impact

The main type of broader impact caused by our work is the reduction of time and money due to the ability to continuously monitor data and hence make critical decisions early. In Appendix A, we provide four prototypical examples of such situations. In Example A, every phone call requires time, thus using up money as well, and if we can accurately quantify uncertainty then we can stop collecting data sooner. In Example B, randomization tests such as those involving permutations are a common way to quantify statistical significance, but they are computationally intensive. Knowing when to stop, based on the test being clearly statistically significant (or clearly far from it), can save on energy costs. In Example D, when an educational intervention is unrolled one school at a time, there are two possibilities again: if it is clearly beneficial, we would like to recognize it quickly so that every student can avail of the benefits, while if it is for some reason harmful (e.g. causing stress without measurable benefit), then it would be equally important to end the program quickly. Once more, accurately quantifying uncertainty as the process unfolds underpins the ability to make these decisions early to disseminate benefits rapidly or mitigate harms quickly.

Our techniques are also applicable to auditing elections (checking whether the results are as announced by a manual random recount). 'Risk-limiting audits' [22] constitute an application area that we intend to pursue; there are many variants depending on how voters express their preferences (choose one, or rank all, or score all) and the aggregation mechanism used to decide on one or multiple winners. Audits are not currently required by law in many elections due to high perceived effort among other reasons, so being able to stop these audits early, yet accurately and confidently, is critical to their broad adoption. Thus, a longer-term broader impact to trust in elections is anticipated.

## Footnotes

[2]Code to reproduce plots is available at github.com/wannabesmith/confseq_wor.

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
