[Supplementary Material]

# A  Four prototypical examples

The following examples are meant to demonstrate situations where we might care about sequentially quantifying uncertainty for parameters of finite populations (see Figure 6).

## A. Opinion surveys (discrete categorical)

Imagine you have access to a registry of phone numbers of a group of 1000 people, such as all residents of a neighborhood, voters in a township, or occupants of a university building. You wish to quickly determine the majority opinion on a categorical question, like preference of Biden vs. Trump. You pick names uniformly at random, call and ask. Obviously, you never call the same person twice. When can you confidently stop? In a typical run on a hypothetical ground truth of 650/350, our method stopped after 123 calls (Figure 6A).

In the example of opinion surveys, the data are discrete and consist of 650 responses showing preference for Biden and 350 showing preference for Trump (encoded as ones and zeros, respectively). The observed data is thus a random permutation of 650 ones and 350 zeros. The CS used was the PPR CS for the hypergeometric distribution with a uniform 'working prior' (i.e. $a = b = 1$ in the beta-binomial pmf).

## B. Permutation $p$-values (discrete binary)

Statistical inference is often performed using permutation tests. Formally, the permutation $p$-value is defined as $P_{\text{perm}} := \frac{1}{m!} \sum_{\pi \in S_m} I(T_m \geqslant T_{\pi(m)})$, where $T_m, T_{\pi(m)}$ are the original and permuted test statistics on $m$ datapoints, and $S_m$ is the set of all $m$-permutations (size $N = m!$). $P_{\text{perm}}$ is intractable to calculate for large $m$, so it is often approximated by randomly sampling $\pi$ with replacement (often 1000 times, fixed and arbitrary). Instead, our tools allow a user to construct a CS for $P_{\text{perm}}$ and sequentially sample WoR until the CS is confident about whether $P_{\text{perm}}$ is below or above (say) 0.05. In one example (small, so we can calculate $P_{\text{perm}} = 0.04$ to verify accuracy), we stopped after 876 steps (Figure 6B).

The permutation test used in this example is a slight modification of the famous 'Lady Tasting Tea' experiment [23]. The experiment proceeds as follows.

There are 12 cups of tea with milk, half of which had the tea poured first, and the other half had milk poured first. The tea expert is told that half of the cups are milk-first and the other half are tea-first and is tasked with determining which ones are which. The null hypothesis is that the tea expert has no ability to distinguish between tea-first and milk-first (i.e. their guesses are independent of the order of milk/tea). Suppose they guess 10 out of 12 cups correctly. The statistical question becomes, "what is the probability of guessing 10 or more cups correctly if the expert is guessing randomly?". This probability is exactly the permutation $p$-value that the statistician is interested in.

To calculate this permutation $p$-value, we consider the set of all possible random guesses that the tea expert could have made, and compute the fraction of those which identify 10 or more cups correctly. If we randomly sample a sequence of possible guesses from the set of $\binom{12}{6}$ possible guesses and record whether 10 or more cups are correctly identified, then observations are a random stream of ones and zeros. We then construct a PPR CS with a uniform 'working prior' for the number of ones, $N^+$ in this set to arrive at a CS for the permutation $p$-value, $P := \frac{N^+}{\binom{12}{6}}$.

## C. Shapley values (bounded real-valued)

First developed in game theory, Shapley values have been recently proposed as a measure of variable or data-point importance for supervised learning. Given a set of players $\{1, \ldots, B\}$ and a reward function $\nu$, the Shapley value $\phi_b$ for player $b$ can be written as an average of $B!$ function evaluations, one for each permutation of $\{1, \ldots, B\}$. As above, $\phi_b$ is intractable to compute and Monte-Carlo techniques are popular. This real-valued setting requires different CS techniques from the categorical setting. As Figure 6C unfolds from left to right (with $B = 7$), it can be stopped adaptively with valid confidence bounds on all $\{\phi_b\}_{b=1}^B$. In this example, we consider a simple cost allocation problem. Suppose there are $n$ people that wish to share transportation to get from point A to their respective destinations, which are all in succession on the same street. Suppose that the cost of going from point A to the $i^{\text{th}}$ person's destination costs $c_i$, and without loss of generality suppose $c_1 < c_2 \cdots < c_n$. In this particular example, we used $n = 7$ with costs of 1, 10, 40, 80, 130, 175, and 200. The 'cost',

$\nu : 2^{[n]} \to \mathbb{R}$ of a trip is defined in the following natural way,

$$
\begin{aligned}
\nu(\varnothing) &= 0 \\
\nu(\{i\}) &= c_i \\
\nu(S) &= c_j \text{ where } c_j \geqslant c_k \text{ for all } k, j \in S
\end{aligned}
$$

The *Shapley value*, $\phi_i$ for person $i$ can be written as,

$$
\phi_i = \frac{1}{n!} \sum_\pi \left[ \nu(S_{\pi,i} \cup \{i\}) - \nu(S_{\pi,i}) \right] \tag{A.1}
$$

where the sum is taken over all permutations $\pi$ of $[n]$, and $S_{\pi,i}$ is the set of numbers to the left of $i$ in the permutation $\pi([n])$.

Since the Shapley value $\phi_i$ is an average of $n!$ numbers, it may be tedious to compute for large $n$ especially when $\nu$ cannot be computed quickly. In our case, the summands have a crude upper bound of $c_n$ and a lower bound of 0 so we can randomly sample WoR from the set of permutations on $[n]$ to construct the empirical Bernstein CS of Theorem 3.2 with the $\lambda$-sequence of (3.13). After 1252 permutations, we are able to conclude with high confidence which player has the highest Shapley value.

## D. Tracking interventions (bounded real-valued)

Suppose a state school board is interested in introducing a new program to help students improve their standardized testing skills. Before deploying it to each of their 3000 public schools, the board decides to incrementally introduce the program to randomly selected schools, measuring standardized test scores before and after its introduction. The board can construct a CS for the overall percentage increase in test scores (which could get worse), and stop the experiment once they are confident about the program's effectiveness. In Figure 6D, with effect size 20%, the board can confidently decide to mandate the program statewide after 260 random schools have been trialed, but they may also continue tracking progress and stop later. In this example, we simply generated 3000 observations from a Beta(3, 2) distribution, appropriately scaled to be between -100 and 100 (representing percentage changes in test scores). To construct a CS for the average change in test scores, we used the Hoeffding-type CS optimized for times 10, 100, and 1000. Note that this CS would be tighter if the empirical Bernstein CS were used as the Beta(3, 2) has a relatively small variance.

Figure 6: Typical simulation runs for the aforementioned examples, with more details in the Supplement. All experiments can be proactively monitored, optionally continued and adaptively stopped.

# B  Proofs of the main results

## B.1  Proof of Proposition 2.1

The proof is broken into two steps. First, we prove that with respect to the filtration $(\mathcal{F}_t)_{t=0}^{N}$ outlined in Section 1.1, the prior-posterior ratio (PPR) evaluated at the true $\theta^\star \in \Theta$,

$$R_t(\theta^\star) := \frac{\pi_0(\theta^\star)}{\pi_t(\theta^\star)}, \tag{B.1}$$

is a nonnegative martingale with initial value one. Later, we invoke Ville's inequality [24, 12] for nonnegative supermartingales to construct the CS.

**Step 1.** Let $\pi_0$ be any prior on $\Theta$ that assigns nonzero mass everywhere. Define the prior-posterior ratio, $R_t(\theta)$ as in (B.1). Writing the conditional expectation of $R_{t+1}(\theta^\star)$ given $X_1^t$ for any $t \in \{1, \ldots, N\}$ in its integral form,

$$
\begin{aligned}
\mathbb{E}(R_{t+1}(\theta^\star) \mid X_1^t) &= \int_{\mathcal{X}_{t+1}} \frac{\pi_0(\theta^\star)}{\pi_{t+1}(\theta^\star)} p_{\theta^\star}(x_{t+1} \mid X_1^t) dx_{t+1} \\
&= \int_{\mathcal{X}_{t+1}} \frac{\pi_0(\theta^\star) \int_{\Theta} p_\eta(X_1^t, x_{t+1}) \pi_0(\eta) d\eta}{p_{\theta^\star}(X_1^t, x_{t+1}) \pi_0(\theta^\star)} p_{\theta^\star}(x_{t+1}) \mid X_1^t) dx_{t+1} && \text{(Bayes' rule)} \\
&= \int_{\mathcal{X}_{t+1}} \frac{\pi_0(\theta^\star) \int_{\Theta} p_\eta(X_1^t, x_{t+1}) \pi_0(\eta) d\eta}{p_{\theta^\star}(X_1^t) \pi_0(\theta^\star)} dx_{t+1} && \text{(Bayes' rule again)} \\
&= \int_{\mathcal{X}_{t+1}} \frac{\pi_0(\theta^\star) \int_{\Theta} p_\eta(X_1^t, x_{t+1}) \pi_0(\eta) d\eta}{\pi_t(\theta^\star) \int_{\Theta} p_\lambda(X_1^t) \pi_0(\lambda) d\lambda} dx_{t+1} && \text{(Bayes' rule again)} \\
&= \frac{\pi_0(\theta^\star)}{\pi_t(\theta^\star)} \int_{\mathcal{X}_{t+1}} \frac{\int_{\Theta} p_\eta(X_1^t, x_{t+1}) \pi_0(\eta) d\eta}{\int_{\Theta} p_\lambda(X_1^t) \pi_0(\lambda) d\lambda} dx_{t+1} \\
&= \frac{\pi_0(\theta^\star)}{\pi_t(\theta^\star)} \frac{\int_{\Theta} \int_{\mathcal{X}_{t+1}} p_\eta(X_1^t, x_{t+1}) dx_{t+1} \pi_0(\eta) d\eta}{\int_{\Theta} p_\lambda(X_1^t) \pi_0(\lambda) d\lambda} && \text{(Fubini's theorem)} \\
&= \frac{\pi_0(\theta^\star)}{\pi_t(\theta^\star)} \frac{\int_{\Theta} p_\eta(X_1^t) \pi_0(\eta) d\eta}{\int_{\Theta} p_\lambda(X_1^t) \pi_0(\lambda) d\lambda} \quad = R_t(\theta^\star).
\end{aligned}
$$

Furthermore, for the case when $t = 0$,

$$
\begin{aligned}
\mathbb{E}(R_1(\theta^\star)) &= \int_{\mathcal{X}_1} \frac{\pi_0(\theta^\star) \int_{\Theta} p_\eta(X_1) \pi_0(\eta) d\eta}{p_{\theta^\star}(X_1) \pi_0(\theta^\star)} p_{\theta^\star}(X_1) dx_1 \\
&= \frac{\pi_0(\theta^\star)}{\pi_0(\theta^\star)} \int_{\mathcal{X}_1} \int_{\Theta} p_\eta(X_1) \pi_0(\eta) d\eta dx_1 && \text{(Bayes' rule)} \\
&= \frac{\pi_0(\theta^\star)}{\pi_0(\theta^\star)} \int_{\Theta} \int_{\mathcal{X}_1} p_\eta(X_1) dx_1 \pi_0(\eta) d\eta && \text{(Fubini's theorem)} \\
&= \frac{\pi_0(\theta^\star)}{\pi_0(\theta^\star)} \int_{\Theta} \pi_0(\eta) d\eta \quad = \frac{\pi_0(\theta^\star)}{\pi_0(\theta^\star)} = R_0 = 1.
\end{aligned}
$$

Establishing that $R_t(\theta^\star)$ is a nonnegative martingale with initial value one completes the first step.

**Step 2.** Ville's inequality for nonnegative supermartingales [24, 12] implies that for any $\beta > 0$,

$$\Pr\left(\exists t \in [N] : R_t(\theta^\star) \geqslant \beta\right) \leqslant \frac{\mathbb{E}(R_0(\theta^\star))}{\beta}.$$

In particular, for a threshold $\alpha \in (0, 1)$,

$$\Pr\left(\exists t \in [N] : R_t(\theta^\star) \geqslant 1/\alpha\right) \leqslant \alpha. \tag{B.2}$$

Define the sequence of sets for $t \in [N]$,

$$C_t := \{\theta : R_t(\theta) \leqslant 1/\alpha\}.$$

As a consequence of (B.2), we have that

$$\Pr\left(\forall t \in [N], \ \theta^\star \in C_t\right) \geqslant 1 - \alpha,$$

as desired, which completes the proof.

## B.2 Proof of Theorem 3.1

*Proof.* Similar to the proof of Proposition 2.1, we proceed in two steps. First, we show that the exponential Hoeffding-type process (3.4) is a nonnegative supermartingale with respect to the filtration outlined in Section 1.1. We then apply Ville's inequality to this supermartingale and ultimately obtain the bound stated in the theorem.

We prove the bound for $[0, 1]$-bounded random variables but the general result holds by taking any $[\ell, u]$-bounded random variable, $X_i$ and applying the transformation, $X_i \mapsto (X_i - \ell)/(u - \ell)$

**Step 1.** Let $(\mathcal{F}_t)_{t=0}^N$ be the filtration defined in Section 1.1. Furthermore, let $\lambda_t \equiv \lambda_t(X_1, \ldots, X_{t-1})$ be a sequence of $\mathcal{F}_{t-1}$-measurable random variables. Consider the exponential Hoeffding-type process $(M_t^H)_{t=0}^N$ with a 'predictable mixture',

$$
M_t^H := \exp\left\{ \sum_{i=1}^{t} \left[ \lambda_i \left( X_i - \mu + Z_{i-1}^\star \right) - \frac{\lambda_i^2}{8} \right] \right\} \equiv \prod_{i=1}^{t} \exp\left\{ \lambda_i \left( X_i - \mu + Z_{i-1}^\star \right) - \frac{\lambda_i^2}{8} \right\}
$$

where $Z_i^\star = \frac{1}{N-i} \sum_{j=1}^{i} (X_j - \mu)$ and $M_0^H = 0$ by convention. Writing the conditional expectation of this process for any $t \geqslant 1$,

$$
\mathbb{E}(M_{t+1}^H \mid \mathcal{F}_t) = \mathbb{E}\left( \prod_{i=1}^{t+1} \exp\left\{ \lambda_i (X_i - \mu + Z_{i-1}^\star) - \frac{\lambda_i^2}{8} \right\} \,\Big|\, \mathcal{F}_t \right)
$$

$$
= M_t^H \cdot \mathbb{E}\left( \exp\left\{ \lambda_{t+1}(X_{t+1} - \mu + Z_t^\star) - \frac{\lambda_{t+1}^2}{8} \right\} \,\Big|\, \mathcal{F}_t \right).
$$

Using the fact that $\mathbb{E}(X_{t+1} - \mu + Z_t^\star \mid \mathcal{F}_t) = 0$, the fact that $X_{t+1} \in [0, 1]$, and that $\lambda_{t+1}$ is $\mathcal{F}_t$-measurable, we have by sub-Gaussianity of bounded random variables,

$$
\mathbb{E}\left( \exp\{\lambda_{t+1}(X_{t+1} - \mu + Z_t^\star)\} \,\Big|\, \mathcal{F}_t \right) \leqslant \exp\left\{ \frac{\lambda_{t+1}^2}{8} \right\}
$$

and thus $\mathbb{E}(M_{t+1}^H \mid \mathcal{F}_t) \leqslant M_t^H$. Therefore, with respect to the filtration $(\mathcal{F}_t)_{t=0}^N$, we have that $M_t^H$ is a nonnegative supermartingale.

**Step 2.** Now that we have shown that $M_t^H$ is a nonnegative supermartingale, we may apply Ville's inequality to obtain,

$$
\Pr\left( \exists t \in [N] : M_t^H \geqslant \frac{1}{\alpha} \right) \leqslant \alpha.
$$

In particular, with probability at least $(1 - \alpha)$, we have that for all $t \in [N]$, $M_t^H < \frac{1}{\alpha}$.

**Step 3.** 'Inverting' the above statement and solving for $\widehat{\mu}_t(\lambda_1^t) - \mu$, we get that with probability at least $(1 - \alpha)$, for all $t \in [N]$,

$$
\widehat{\mu}_t(\lambda_1^t) - \mu < \frac{\sum_{i=1}^{t} \lambda_i^2/8 + \log(1/\alpha)}{\sum_{i=1}^{t} \lambda_i \left( 1 + \frac{i-1}{N-i+1} \right)}.
$$

Applying all of the aforementioned logic to $-X_1, \ldots, -X_t$ and $-\mu$, and taking a union bound, we arrive at the desired result,

$$
\Pr\left( \exists t \in [N] : |\widehat{\mu}_t(\lambda_1^t) - \mu| \geqslant \frac{\sum_{i=1}^{t} \lambda_i^2/8 + \log(2/\alpha)}{\sum_{i=1}^{t} \lambda_i \left( 1 + \frac{i-1}{N-i+1} \right)} \right) \leqslant \alpha,
$$

which completes the proof.

$\square$

**Remark:** $\widehat{\mu}_t$ **is unconditionally unbiased.** Recalling the advantage term $A_t := \sum_{i=1}^{t} \frac{i-1}{N-i+1}$, a short calculation shows that $\widehat{\mu}_t$ (3.1) has conditional expectation equaling a convex combination of $\widehat{\mu}_t, \mu$:

$$
\mathbb{E}[\widehat{\mu}_{t+1}|X_1^t] = \frac{1 + A_{t+1} - A_t}{t + 1 + A_{t+1}} \mu + \frac{t + A_t}{t + 1 + A_{t+1}} \widehat{\mu}_t.
$$

Multiplying both sides by $t + 1 + A_{t+1}$, we can write it in a recursive, telescoping form:
$$\mathbb{E}[(t + 1 + A_{t+1})\widehat{\mu}_{t+1}|X_1^t] = \mu + (A_{t+1} - A_t)\mu + (t + A_t)\widehat{\mu}_t.$$
Taking expectation with respect to $X_t|X_1^{t-1}$, and using the above equation to evaluate the last term,
$$\mathbb{E}[(t + 1 + A_{t+1})\widehat{\mu}_{t+1}|X_1^{t-1}] = 2\mu + (A_{t+1} - A_{t-1})\mu + (t - 1 + A_{t-1})\widehat{\mu}_{t-1}.$$
Unrolling this process out, we see that $\mathbb{E}[(t + 1 + A_{t+1})\widehat{\mu}_{t+1}] = (t + 1)\mu + (A_{t+1} - A_0)\mu$. Since $A_0 \equiv 0$, we conclude that $\widehat{\mu}_{t+1}$ is an unconditionally unbiased estimator of $\mu$.

Interestingly, the without-replacement mean estimator is not necessarily 'consistent' (in the sense of recovering $\mu$ after all $N$ samples are drawn). However, the concept of consistency is subtle for finite populations as there is no longer any uncertainty after all samples are drawn. In any case, the without-replacement mean estimator was not introduced to replace the usual sample mean estimator in all without-replacement settings, but was simply the quantity that resulted from attempting to develop exponential supermartingales within this sample scheme.

### B.3 Proof of Theorem 3.2

*Proof.* Much like the proof of Theorem 3.1, the proof proceeds in three steps: (1) showing that an exponential empirical Bernstein-type process is a supermartingale, (2) applying Ville's inequality, and (3) inverting the process and taking a union bound. Again, we prove the result for $[0, 1]$-bounded random variables since for an $[\ell, u]$-bounded random variable $X_i$, one can make the transformation $X_i \mapsto (X_i - \ell)/(u - \ell)$

**Step 1.** Let $(\mathcal{F}_t)_{t=0}^N$ be the filtration defined in Section 1.1. Let $\lambda_t \equiv \lambda_t(X_1, \ldots, X_{t-1})$ be a sequence of $\mathcal{F}_{t-1}$-measurable random variables. Consider the exponential empirical Bernstein-type process, $(M_t^E)_{t=0}^N$ with a 'predictable mixture',

$$M_t^E := \exp\left\{\sum_{i=1}^t \left[\lambda_i \left(X_i - \mu + Z_{i-1}^{\star}\right) - 4(X_i - \widehat{\mu}_{i-1})^2 \psi_E(\lambda_i)\right]\right\}$$

$$\equiv \prod_{i=1}^t \exp\left\{\lambda_i \left(X_i - \mu + Z_{i-1}^{\star}\right) - 4(X_i - \widehat{\mu}_{i-1})^2 \psi_E(\lambda_i)\right\}$$

where $M_0^E := 0$. Writing out the conditional expectation of $M_{t+1}^E$ given $\mathcal{F}_t$ for $t \in [N]$,

$$\mathbb{E}\left(M_{t+1}^E \mid \mathcal{F}_t\right) = M_t^E \cdot \mathbb{E}\left(\exp\left\{\lambda_{t+1}\left(X_{t+1} - \mu + Z_t^{\star}\right) - 4\psi_E(\lambda_{t+1})\left(X_{t+1} - \widehat{\mu}_t\right)^2\right\} \;\middle|\; \mathcal{F}_t\right).$$

Therefore, it suffices to show that for any $t \in [N]$,

$$\mathbb{E}\left(\exp\left\{\lambda_{t+1}\left(X_{t+1} - \mu + Z_t^{\star}\right) - 4\psi_E(\lambda_{t+1})\left(X_{t+1} - \widehat{\mu}_t\right)^2\right\} \;\middle|\; \mathcal{F}_t\right) \leqslant 1.$$

For succinctness, denote

$$Y_{t+1} := X_{t+1} + \frac{1}{N - t}\sum_{j=1}^t X_j - \frac{N}{N - t}\mu \quad \text{and} \quad \delta_t := \widehat{\mu}_t + \frac{1}{N - t}\sum_{j=1}^t X_j - \frac{N}{N - t}\mu.$$

Note that $Y_{t+1}$ is conditionally mean zero. It then suffices to prove that for any $(0, 1)$-bounded, $\mathcal{F}_t$-measurable $\lambda_{t+1} \equiv \lambda_{t+1}(X_1, \ldots, X_t)$,

$$\mathbb{E}\left[\exp\left\{\lambda_{t+1}Y_{t+1} - 4(Y_{t+1} - \delta_t)^2\psi_E(\lambda_{t+1})\right\} \;\middle|\; \mathcal{F}_t\right] \leqslant 1.$$

Indeed, in the proof of Proposition 4.1 in Fan et al. [25], $\exp\{\xi\lambda - 4\xi^2\psi_E(\lambda)\} \leqslant 1 + \xi\lambda$ for any $\lambda \in [0, 1)$ and $\xi \geqslant -1$. Setting $\xi := Y_{t+1} - \delta_t = X_{t+1} - \widehat{\mu}_t$,

$$\mathbb{E}\left[\exp\left\{\lambda_{t+1}Y_{t+1} - 4(Y_{t+1} - \delta_t)^2\psi_E(\lambda_{t+1})\right\} \;\middle|\; \mathcal{F}_t\right]$$

$$= \mathbb{E}\left[\exp\left\{\lambda_{t+1}(Y_{t+1} - \delta_t) - 4(Y_{t+1} - \delta_t)^2\psi_E(\lambda_{t+1})\right\} \mid \mathcal{F}_t\right]\exp(\lambda_{t+1}\delta_t)$$

$$\leqslant \mathbb{E}\left[1 + (Y_{t+1} - \delta_t)\lambda_{t+1} \mid \mathcal{F}_t\right]\exp(\lambda_{t+1}\delta_t) \overset{(i)}{=} \mathbb{E}\left[1 - \delta_t\lambda_{t+1} \mid \mathcal{F}_t\right]\exp(\lambda_{t+1}\delta_t) \overset{(ii)}{\leqslant} 1,$$

where equality $(i)$ follows from the fact that $Y_{t+1}$ is conditionally mean zero as mentioned earlier, and inequality $(ii)$ follows from the inequality $1 - x \leqslant \exp(-x)$ for all $x \in \mathbb{R}$.

**Step 2.** Now that we have established that $M_t^E$ is a nonnegative supermartingale, we apply Ville's inequality to obtain,

$$\Pr\left(\exists t \in [N] : M_t^E \geqslant \frac{1}{\alpha}\right) \leqslant \alpha.$$

**Step 3.** Solving for $\widehat{\mu}_t - \mu$ in the inequality in the above probability statement, we get that

$$\Pr\left(\exists t \in [N] : \widehat{\mu}_t - \mu \geqslant \frac{\sum_{i=1}^t 4\psi_E(\lambda_i)(X_i - \widehat{\mu}_{i-1})^2 + \log(1/\alpha)}{\sum_{i=1}^t \lambda_i \left(1 + \frac{i-1}{N-i+1}\right)}\right) \leqslant \alpha.$$

Applying the same logic to $-X_1, \ldots, -X_t$ and $-\mu$, and taking a union bound, we arrive at the desired result,

$$\Pr\left(\exists t \in [N] : |\widehat{\mu}_t - \mu| \geqslant \frac{\sum_{i=1}^t 4\psi_E(\lambda_i)(X_i - \widehat{\mu}_{i-1})^2 + \log(2/\alpha)}{\sum_{i=1}^t \lambda_i \left(1 + \frac{i-1}{N-i+1}\right)}\right) \leqslant \alpha.$$

$\square$

# C   Sampling multivariate binary variables WoR

The prior-posterior martingale from Section 2.2 extends naturally to the multivariate case as follows. Suppose we have $N$ objects, each belonging to one of $K \geqslant 2$ categories, and there are $N_1^\star, \ldots, N_K^\star$ objects from each category, respectively. Let $c$ denote the category of a randomly sampled object, and let

$$\mathbf{X} := \begin{pmatrix} \mathbb{1}(c = 1) & \mathbb{1}(c = 2) & \cdots & \mathbb{1}(c = K) \end{pmatrix}.$$

Then $\mathbf{X}$ is said to follow a multivariate hypergeometric distribution with parameters $N$, $(N_1^\star, \ldots, N_K^\star)$, and $n = 1$ and has probability mass function,

$$\Pr(\mathbf{X} = x) = \frac{\prod_{k=1}^K \binom{N_k^\star}{x_k}}{\binom{N}{n}}.$$

Note that $\sum_{k=1}^K x_k = 1$ and $x_k \in \{0, 1\}$ for each $k \in \{1, \ldots, K\}$. More generally, if $n \geqslant 2$ objects are sampled WoR, then $\mathbf{X}$ would have the same probability mass function with $x_1, \ldots, x_K \in \{1, \ldots, n\}$ such that $\sum_{k=1}^K x_k = n$. As in Section 2.2, we will consider the case where $n = 1$ for notational simplicity.

Let us now view this random variable and the fixed multivariate parameter $\mathbf{N}^\star := (N_1^\star, \ldots, N_K^\star)$ from the Bayesian perspective as in Section 2.2 by treating $\mathbf{N}^\star$ as a random variable which we denote by $\widetilde{\mathbf{N}}^\star$ to avoid confusion. Suppose that

$$\mathbf{X}_t \mid (\widetilde{\mathbf{N}}^\star, \mathbf{X}_1, \ldots, \mathbf{X}_{t-1}) \sim \text{MultHyperGeo}\left(N - (t-1), \widetilde{\mathbf{N}}^\star - \sum_{i=1}^{t-1} \mathbf{X}_i, 1\right), \quad \text{and}$$

$$\widetilde{\mathbf{N}}^\star \sim \text{DirMult}(N, \mathbf{a})$$

for some $\mathbf{a} := (a_1, \ldots, a_K)$ with $a_k > 0$ for each $k \in \{1, \ldots, K\}$. Then for any $t \in \{1, 2, \ldots, N\}$,

$$\widetilde{\mathbf{N}}^\star - \sum_{i=1}^t \mathbf{X}_i \mid (\mathbf{X}_1, \ldots, \mathbf{X}_t) \sim \text{DirMult}\left(N - t, \mathbf{a} + \sum_{i=1}^t \mathbf{X}_i\right).$$

With these prior and posterior distributions, we're ready to invoke Proposition 2.1 to obtain a sequence of confidence sets for $\mathbf{N}^\star$.

**Theorem C.1** (Confidence sequences for multivariate hypergeometric parameters). *Suppose that*

$$\mathbf{X}_t \mid (\mathbf{X}_1, \ldots, \mathbf{X}_{t-1}) \sim \textit{MultHyperGeo}\left(N - (t-1), \mathbf{N}^\star - \sum_{i=1}^{t-1} \mathbf{X}_i, 1\right).$$

*Let $\pi_0$ and $\pi_t$ be the Dirichlet-multinomial prior with positive parameters $\mathbf{a} = (a_1, \ldots, a_K)$ and corresponding posterior, $\pi_t$, respectively. Then the sequence of sets $(C_t)_{t=0}^N$ defined by*

$$C_t := \left\{ \mathbf{n} \in \{0, \ldots, N\}^K : \sum_{k=1}^{K} \mathbf{n}_k = N \text{ and } \frac{\pi_0(\mathbf{n})}{\pi_t(\mathbf{n})} < \frac{1}{\alpha} \right\}$$

*is a $(1-\alpha)$-CS for $\mathbf{N}^\star$. Furthermore, the running intersection, $\bigcap_{s \leqslant t} C_t$ is a $(1-\alpha)$-CS for $\mathbf{N}^\star$.*

*Proof.* This is a direct consequence of Theorem 2.1 applied to the multivariate hypergeometric distribution with a Dirichlet-multinomial prior. $\square$

## D    Coupling the 'prior' with the stopping rule to improve power

Somewhat at odds with their intended use-case, working 'priors' need not always be chosen to reflect the user's prior information. When approximating $p$-values for permutation tests, for example, it is of primary interest to conclude whether $P_{\text{perm}}$ is above or below some prespecified $\alpha_{\text{perm}} \in (0, 1)$ with high confidence as quickly as possible. As discussed in Theorem 2.1, the CS for $P_{\text{perm}}$ will shrink to a single point regardless of the prior, so if $P_{\text{perm}}$ is much larger or much smaller than $\alpha_{\text{perm}}$, we expect to discover the decision rule, "reject" versus "do not reject" rather quickly. It is when $P_{\text{perm}}$ is very close to $\alpha_{\text{perm}}$ that the user desires sharper confidence intervals, so that they can make decisions sooner (see Figure 7). In this case, they simply need to place more mass near the decision boundary, with a necessary tradeoff between the sharpness of confidence sets near $\alpha_{\text{perm}}$ and the size of the neighborhood around $\alpha_{\text{perm}}$ for which this sharpness is realized.

Figure 7: Comparing priors for Example B: using a uniform prior versus a prior peaked near 0.05. When the decision rule is to stop whenever the CS is entirely on one side of 0.05, coupling the prior to the decision rule leads to earlier stopping.

## E    Choosing a $\lambda$-sequence for Hoeffding and empirical Bernstein CSs

Recall the Hoeffding-type CS of Theorem 3.1,

$$C_t^H := \widehat{\mu}_t(\lambda_1^t) \pm \underbrace{\frac{\sum_{i=1}^{t} \psi_H(\lambda_i) + \log(2/\alpha)}{\sum_i \lambda_i \left(1 + \frac{i-1}{N-i+1}\right)}}_{\text{width } W_t}$$

In Section 3, we presented the $\lambda$-sequence,

$$\lambda_t := \sqrt{\frac{8 \log(2/\alpha)}{t \log(t+1)(u-\ell)^2}} \wedge \frac{1}{u - \ell}. \tag{E.1}$$

This is visually similar to the single value of $\lambda \in \mathbb{R}$,

$$\lambda := \sqrt{\frac{8 \log(2/\alpha)}{t_0(u-\ell)^2}}$$

which optimizes the bound for time $t_0$. Two natural questions arise: (1) where did the extra $\log(t)$ in (E.1) come from, and (2) why this particular $\lambda$-sequence and not others? The answers to these questions are based on some heuristics derived by Waudby-Smith and Ramdas [26] in the with-replacement setting. To make matters simpler, ignore the $\left(1 + \frac{i-1}{N-i+1}\right)$ term in the CS and consider the scaling of the width $W_t$,

$$W_t \asymp \frac{\sum_{i=1}^{t} \psi_H(\lambda_i)}{\sum_{i=1}^{t} \lambda_i} \asymp \frac{\sum_{i=1}^{t} \lambda_i^2}{\sum_{i=1}^{t} \lambda_i}.$$

When the method of mixtures is used to obtain CSs in the with-replacement setting, their widths often follow a $\sqrt{\log t / t}$ rate [1]. Following the approximations in Table 1, we may opt to pick a sequence $(\lambda_i)_{i=1}^{\infty}$ which scales like $1/\sqrt{i \log i}$ to obtain a width $W_t \asymp \sqrt{\log t / t}$. In particular, scaling $\lambda_i$ as $1/\sqrt{i \log i}$ is simply an effort to obtain CSs with reasonable widths. The same arguments combined with (3.9) can be applied to the empirical Bernstein CS to obtain (3.13).

Furthermore, we truncate the $\lambda$-sequence in E.1 to prevent the CS width from being dominated by large $\lambda_t$ at small $t$. It is important to keep in mind that *any* sequence would have yielded a valid CS. The choice presented here was derived based on a heuristic argument and kept because of its reasonable empirical performance.

| Sequence $(\lambda_i)_{i=1}^{\infty}$ | $\sum_{i=1}^{t} \lambda_i$ | $\sum_{i=1}^{t} \lambda_i^2$ | Width $W_t$ |
|---|---|---|---|
| $\asymp 1/i$ | $\asymp \log t$ | $\asymp 1$ | $1/\log t$ |
| $\asymp \sqrt{\log i}/i$ | $\asymp \sqrt{t \log t}$ | $\asymp \log^2 t$ | $\asymp \log^{3/2} t / \sqrt{t}$ |
| $\asymp 1/\sqrt{i}$ | $\asymp \sqrt{t}$ | $\asymp \log t$ | $\asymp \log t / \sqrt{t}$ |
| $\asymp 1/\sqrt{i \log i}$ | $\asymp \sqrt{t/\log t}$ | $\asymp \log \log t$ | $\asymp \sqrt{\log t / t}$ |
| $\asymp 1/\sqrt{i \log i \log \log i}$ | $\asymp \sqrt{t/\log t}$ | $\asymp \log \log \log t$ | $\asymp \sqrt{\log t / t}$ |

Table 1: Above, we think of $\log x$ as $1 \vee \log(1 \vee x)$ to avoid trivialities. The claimed rates are easily checked by approximating the sums as integrals, and taking derivatives. For example, $\frac{d}{dx} \log \log x = 1/x \log x$, so the sum of $\sum_{i \leqslant t} 1/i \log i \asymp \log \log t$. It is worth remarking that for $t = 10^{80}$, the number of atoms in the universe, $\log \log t \approx 5.2$, which is why we treat $\log \log t$ as a constant when expressing the rate for $W_t$. The iterated logarithm pattern in the the last two lines of the table can be continued indefinitely.

# F  Comparing our CSs to those implied by Bardenet & Maillard

Bardenet & Maillard [14, Theorem 2.4] provide the following two time-uniform Hoeffding-Serfling inequalities when sampling bounded real numbers WoR from a finite population. For any $n \in [N]$,

$$\Pr\left(\exists t \in \{1, \ldots, n\} : \frac{1}{N-t} \sum_{i=1}^{t} (X_i - \mu) \geqslant \frac{n\epsilon}{N-n}\right) \leqslant \exp\left\{-\frac{2n\epsilon^2}{(1-(n-1)/N)(u-\ell)^2}\right\} \quad \text{and}$$

$$\Pr\left(\exists t \in \{n, \ldots, N-1\} : \frac{1}{t} \sum_{i=1}^{t} (X_i - \mu) \geqslant \epsilon\right) \leqslant \exp\left\{-\frac{2n\epsilon^2}{(1-n/N)(1+1/n)(u-\ell)^2}\right\}.$$

Inverting these inequalities and taking a union bound to get two-sided inequalities, we have

$$\frac{1}{t} \sum_{i=1}^{t} X_i \pm \frac{n(N-t)}{t(N-n)} \sqrt{\frac{\log(4/\alpha)(1-(n-1)/N)(u-\ell)^2}{2n}} \qquad \text{when } t \leqslant n \qquad \text{(F.1)}$$

$$\frac{1}{t} \sum_{i=1}^{t} X_i \pm \sqrt{\frac{\log(4/\alpha)(1-n/N)(1+1/n)(u-\ell)^2}{2n}} \qquad \text{when } t \geqslant n \qquad \text{(F.2)}$$

is a $(1-\alpha)$ CS for $\mu$. We term the CS defined by (F.1) and (F.2) as the Bardenet-Maillard CS for simplicity.

A comparison of the aforementioned CS to our Hoeffding-type CS is displayed in Figure 9, where we see that our bound is roughly as tight as the Bardenet-Maillard CS at the time of optimization,

Figure 8: Hoeffding CSs based on fixed $\lambda$ values optimized for times 30 and 800, respectively alongside the CS based on the $\lambda$-sequence in (E.1). Notice that no CS uniformly dominates the others, but that the sequence in (E.1) acts as a middle ground between the other two.

while our bounds are (much) tighter everywhere else. This phenomenon was observed and studied in the with-replacement setting, attributing the benefits of confidence bounds like our Hoeffding CS to an underlying 'line-crossing' inequality being uniformly tighter than an underlying Freedman-type inequality. For more information on the with-replacement analogy, we direct the reader to the pair of papers by Howard et al. [1, 12]. Returning back to the WoR setting, we remark that (F.1) uses the standard sample mean, but we use a more sophisticated sample mean (3.1).

Figure 9: A comparison of our Hoeffding-type CS against the Hoeffding-Serfling CS of Bardenet & Maillard [14]. Our Hoeffding CS appears to be as tight as the Hoeffding-Serfling bound at the time of optimization, but tighter at all other times.

# G    Time-uniform versus fixed-time bounds

A natural question to ask is, 'how much does one sacrifice by using a time-uniform CS instead of a fixed-time confidence interval'? The answer to this question will depend largely on the type of bound used, the underlying finite population, and other factors. However, in the case of sampling binary numbers from a finite population, it seems that the answer is 'not much'. In Figure 10, we display the fixed-time Hoeffding confidence interval of Corollary 3.1 alongside its time-uniform counterpart from Theorem 3.1 and the prior-posterior ratio CS from Theorem 2.1. In terms of the width of confidence bounds, we find that not much is lost by using the two aforementioned CSs over the fixed-time Hoeffding confidence interval. For this small price, the user is awarded the flexibility that comes with using CSs such as properties (a), (b), and (c) described in the Introduction.

Figure 10: Comparing fixed-time and time-uniform confidence bounds for sampling binary numbers from a population of size 1000, consisting of 500 ones and 500 zeros. The dotted red line shows the fixed-time Hoeffding bound of Corollary 3.1, while the dashed green and solid blue lines refer to the time-uniform Hoeffding-type CS and the prior-posterior ratio CS, respectively. Notice that the increase in confidence bound width that results from using a time-uniform bound is relatively minor.

# H    Computational considerations

When using the CSs of Theorems 2.1, 3.1, and 3.2 in practice, it is important to keep in mind the computational costs associated with each method. For fixed values of $\lambda$, updating the Hoeffding and empirical Bernstein CSs at a each time $t$ takes constant time and constant memory, since all calculations involve cumulative sums (or averages). Furthermore, optimal values of $\lambda$ can be computed as in (3.6) for Hoeffding-type bounds and approximated as in (3.11) for empirical Bernstein-type bounds, all in constant time. On the other hand, the prior-posterior ratio (PPR) CS of Theorem 2.1 is the more computationally expensive method among those presented, but can still be computed quickly for many problems. In order to find the CS,

$$C_t := \left\{ n^+ \in [N] : \frac{\pi_0(n^+)}{\pi_t(n^+)} < \frac{1}{\alpha} \right\},$$

one must find all values in $\{0, \ldots, N\}$ which, when provided as an input to $\frac{\pi_0(\cdot)}{\pi_t(\cdot)}$ are less than $1/\alpha$.

Therefore, computing the entire CS takes $O(PN^2)$ time where $P$ is the time required to compute $\pi_0(n)/\pi_t(n)$. In all of the PPR CSs presented in this paper, we used computationally tractable conjugate priors, so $P = 1$. We believe more sophisticated root-finding methods can be employed to arrive at a time of $O(N \log(PN))$, but these methods are reasonably fast in our experience. Moreover, the PPR CS can be computed on a subset of $[N]$ if needed, and is parallelizable.

For reference, we provide average computation times in Table 2. All calculations were measured using Python's default `time` package and were performed in Python 3.8.3 using the `numpy` and `scipy` packages on a quad-core CPU with 8 threads at 1.8GHz each. However, no parallel processing was performed aside from the default multithreading provided by Python.

|  | Time in seconds (std. dev.) |
| --- | --- |
| Hoeffding | $2.13 \times 10^{-4}$ ($2.88 \times 10^{-5}$) |
| Empirical Bernstein | $2.35 \times 10^{-4}$ ($3.24 \times 10^{-5}$) |
| Prior-posterior ratio | 0.306 (0.0115) |

Table 2: Average time taken to compute the various CSs for $N = 1000$ discrete observations with equal numbers of ones and zeros, with standard deviations for 100 repeated experiments.

# I   Simple experiments for computing miscoverage rates

Figure 11: Empirical miscoverage probabilities for our empirical Bernstein, Hoeffding, and prior-posterior CSs. The left plot compares empirical Bernstein and Hoeffding for a population of $N = 10,000$ consisting of bounded, real-valued observations uniformly distributed on the unit interval. The plot on the right-hand side compares all three for a population of the same size containing discrete elements with zeros and ones in equal proportions. Notice that while the empirical Bernstein CS does reasonably well in both settings, none of the three methods uniformly dominates the others.

Typically, in nonparametric testing, there is no 'uniformly most powerful' test: any test achieving high power against some class of alternatives must necessarily be less powerful against some other class of alternatives, while a different test may display the opposite behavior. An analogous story holds for nonparametric estimation as well: the class of bounded random variables (or sequences of bounded random numbers) is nonparametric, and in such a setting, no single estimation technique can uniformly dominate all others (that is, always have lower width for any bounded sequence). This phenomenon is easy to exemplify for our confidence sequences: we can construct settings where the Hoeffding-type CS is less conservative (tighter estimation, more powerful as a test) than the empirical-Bernstein CS, and other settings in which the opposite is true. Figure 11 considers two such 'opposite' scenarios: the binary setting which maximizes the variance of the sequence, and another setting in which the observations are uniformly distributed on $[0, 1]$. In the first setting, there is no point in 'estimating' the variance (empirical-Bernstein) as opposed to just assuming that it is the maximum possible variance (Hoeffding-type), and so the former is more conservative than the latter. In the second setting, the Hoeffding CS is far more conservative, as expected. With no prior knowledge on the type of sequence to be encountered, the empirical Bernstein CS seems like a safer choice.