[Reviews · NeurIPS 2020]

Review 1

Summary and Contributions: The paper extends results for sequential hypothesis testing and confidence sequences to the case where data is sampled without replacement. ===== Update: I have read the rebuttal and look forward to seeing the final version of the paper.

Strengths: The paper is presents interesting and novel techniques. It is very well written with a good discussion of issues and choices one faces when applying the methods. While sequential inference problems are relevant to NeurIPS, the problem seems somewhat niche. However, it presents an interesting and potentially high impact possible use case in election auditing.

Weaknesses: The primary weakness is the limited scope of problems which it can offer significant advantages over with replacement sampling. In some cases the work seems to have limited benefit over with replacement sampling. In figure 1D show, the benefits of without replacement sampling for uniform distributions appear to be so slight as to be nearly unreadable from the plot, and one would expect it only appears when the sampled fraction is a large of the population size.

Correctness: Yes, I believe they are. The high level explanations are consistent with the given results though I did not verify the proofs in detail.

Clarity: The paper is very well written. The main suggestion is to reduce the liberal use of quotes and underlining. e.g. "peeking", "p-hacking". I'm not sure why 'filtration' is in quotes in line 80. Is it not a filtration? Underlining the part on 'working priors/posteriors' seems excessive since it's not important to the contributions of the paper.

Relation to Prior Work: Yes

Reproducibility: Yes

Additional Feedback: Overall, the good "science" of the paper makes what will possibly be lasting contributions to the field, not just short term incremental improvements. It would be nice to discuss briefly the advantages/disadvantages of the novel mean estimator in eq 3.1. In particular, the estimator appears to be inconsistent even though it is unbiased since it seems to weight items that occur early more heavily. What are the implications of that? Is there then a point at which without replacement confidence interval is worse than a with replacement confidence interval? details: In Figure 4 (left), it's not clear that the value at t_1=100 is visible on the plot or what the difference in the CIs at that time.


Review 2

Summary and Contributions: The paper provides a novel approach to constructing confidence sequences, i.e., confidence sets (intervals) for parameter estimates that are true simultaneously in all steps of a sequentially refined estimation process. The key idea/result is that if one defines a ‘working prior’ over the parameter space then, at all times, the set of parameters that have a prior/posterior probability ratio of less than 1/alpha form a confidence sequence. Importantly, the choice of the prior does not influence the frequentist validity of the confidence sequence but merely influences its shape. The authors apply this idea to a range of discrete and continuous parameter estimation problems by constructing suitable priors with tractable prior/posterior ratios.

Strengths: The authors attack an extremely important problem that we all face in empirical science where experiments are most naturally performed sequentially and one needs practical but probabilistically sound stopping criteria. Furthermore, the paper and the appendix are full of impressive probabilistic results that are required to apply the core idea to the various settings. Finally, despite the inherent complexity of the topic, the authors have created an accessible exposition and provide well designed didactic application examples.

Weaknesses: I don’t see major weaknesses. See below for a few suggestions for improvement.

Correctness: I did not check the proofs of the main results, which are extensive and only provided in the appendix. However, all the statements are plausible, and the content of the main paper is technically rigorous. Also the authors provide a range of examples which empirically support all key claims, most importantly that the true parameter is contained in the complete confidence sequence, and also that the confidence sequences follow expected shapes (e.g., when comparing the Bernstein versus the Hoeffding style approaches for estimating real-valued parameters).

Clarity: As mentioned above, the paper is excellently written with an accessible introduction to the general problem setting including canonical examples for the different specific settings. At two points I felt some additional details could be provided to further increase the accessibility to non-expert readers: a) The definitions of filtrations and martingale are not recapped. While this is understandable given the space limitations, it constitutes quite a large gap from the intuitive examples. b) In contrast to the other applications, the introduction to Shapley values does not feel self-contained.

Relation to Prior Work: The paper generally does an excellent job in discussing relations to traditional approaches to the construction of confidence sets. A notable potential improvement could be to also include experiments that contrast the confidence sequence approach with traditional confidence bounds that only hold at a single moment in time (i.e., predefined stopping time).

Reproducibility: Yes

Additional Feedback: T_n in page 2, l47 should be T_m UPDATE: I thank the authors for their response to my comments and their intention to use them to further improve the (already very good) presentation of their work.


Review 3

Summary and Contributions: Paper provided CS under sampling without replacement, by a method of PPR martingale. Cases with binary values and real values are discussed. --- I'd like to thank the authors for the discussion and addressing my comments. My score stays as top 50% acceptance.

Strengths: The work addresses important problem of constructing CS under the setup of sampling without replacement. Several imporatant cases are carefully analyzed and corresponding methods, along with associated theory are developed. Paper is very clear and easy to read and the problem is of great interest for researchers in other fields like politics and game theory.

Weaknesses: My only concern is around N - sometime we are sampling without replacement, and without knowing the population size N (e.g. if doing political polls, the analyst usually have a target sample size in mind, say I am gonna sample 1k this time, but the true population under there is not known) Is this method still viable in this case with another layer of randomness? Also some examples are useful in illustrating the type of cases considered, but not quite useful for adapting the method. e.g. permutation test - one can totally do a sampling without replacement (actually that would be easier, just do sampling without replacement on [n] independently), and as n! is usually very large, the difference between CS on sampling with/without replacement would be very small (I am making this claim w/o deriving, but purely on intuition). So another example may be better for selling the paper.

Correctness: I checked the prop 2.1 but not the rest. Prop2.1 is solid.

Clarity: Yes. Clear and easy to read.

Relation to Prior Work: Yes

Reproducibility: Yes

Additional Feedback:


Review 4

Summary and Contributions: This paper develops a variety of confidence sequences for sampling without replacement. In particular, confidence sequences are constructed for categorical observations and the mean of a finite set of bounded real numbers.

Strengths: The main strength is its theoretical contribution, not only the proposed four confidence sequences, but also new technical tools such as prior-posterior-ratio martingale and two additional exponential supermartingales in eq (3.3) and (3.7). These make this paper not a naïve application/extension of existing confidence sequence results, but of its own interest.

Weaknesses: The PPR framework in Section 2 is restricted to the case where the posterior distribution is computable, and the supermartingales setting in Section 3 are designed specifically for the mean of a finite set of bounded real; these restrictions prevent the proposed CS to be used in more general settings.

Correctness: Correct.

Clarity: This paper is well written and well organized.

Relation to Prior Work: Yes.

Reproducibility: Yes

Additional Feedback: The four motivating examples are of limited interest within the machine learning community, and the author might add one or more examples that could have a stronger connection with the community, Moreover, this paper focuses on the discrete categorial setting and a finite set of bounded real, so it might be interesting to add some discussion for more general model and enlarge the applicability of the proposed CS. For the PPR method, the example in sec 2.2 utilize the advantage of a conjugate prior, but that is not always possible; so I am wondering when the posterior distribution is not available in close-form, can we use some sampling method (like MCMC) such that the proposed CS can still be applied and what about the computation complexity? Minor comments: There lacks several right bracket in the equation in step 1 of Appendix A.1. The notation J in Thm 3.1 appears abruptly without any further explanation. -------------------------------------------------------- I have read the author response and I stand by my original score (vote for acceptance).

[Author Response · NeurIPS 2020]

We thank the reviewers for their comments and suggestions which have ultimately helped to improve the paper.

Reviewers 1 and 3 made similar remarks that for large $N$ and small sample sizes, with-replacement (WR) and without-replacement (WoR) confidence sequences (CSs) perform similarly. We fully agree with this sentiment. However, this is a phenomenon surrounding WR and WoR methods in general, and is not limited to our methods. We still see two upsides to using our WoR methods:

- All our confidence sequences shrink to zero width in exactly $N$ steps. With WR sampling, this would not be true at time $N$ (but it would be true asymptotically).
- Given that our new bounds are explicit and simple to implement, there is no reason not take advantage of any available statistical benefit (small at early times, large at later times).

In general, WoR sampling could be utilized on problems with moderate $N$ (e.g. sampling points in stochastic gradient methods, sampling covariates in a random order for coordinate descent, sampling columns of a matrix for accurate approximation, sampling edges in a graph for graph sparsification with spectral approximation, etc). We will point this out on the extra ninth page if the paper is accepted and hope others may follow up on these applications, where we suspect there is a practical benefit to our tighter bounds.

**Reviewer #1**

**(R1) 'It would be nice to discuss [advantages of] the novel mean estimator...'** In the paper, we discuss the advantage (denoted by $A_t$) of the novel mean estimator in terms of its statistical efficiency. However, this novel estimator is used primarily because the resulting Hoeffding- and empirical Bernstein-type processes turn out to be supermartingales, a key property allowing for the derivation of CSs. In some sense, this mean estimator naturally popped out when we were explicitly trying to construct a supermartingale, rather than the other way around.

**(R1) 'The novel mean estimator appears inconsistent'** Consistency is a subtle question; once we have observed $N$ data points, there is no uncertainty left so we do not need an estimator at the final step. Our CSs in fact shrink to zero width at time $N$ because of logical considerations. We will discuss this and better clarify the role of $\widehat{\mu}_t$ in the paper (using the extra space provided if accepted).

**(R1) 'reduce the liberal use of quotes and underlining'** We have now removed most of the quotes and underlines.

**(R1) 'In Figure 4 (left)...'** We now clarify in the caption that time 100 is on the plot.

**Reviewer #2**

**(R2) 'definitions of filtrations and martingales...'** We agree and have now added these.

**(R2) 'introduction to Shapley values not self-contained'** We agree, we have now added additional details for the uninitiated reader.

**(R2) '[...] include experiments [comparing CS to CI]'** Our approach also yields tight fixed-time confidence intervals for sampling WoR. We have plotted these in Figure 9 in Appendix H. We have now added references to these to the main part of the paper, and have added some fixed-time corollaries.

**Reviewer #3**

**(R3) 'My only concern is around N'** It is standard in the mathematical study of WoR sampling to consider the case of known $N$. The case of unknown $N$ is an interesting direction for future work.

**Reviewer #4**

**(R4) The PPR framework...conjugacy/MCMC** We now clarify that the results surrounding the PPR martingale are valid even when the posterior is not in closed-form, and thus methods like MCMC can be applied.

**(R4) 'this paper focuses on the discrete [and] finite set of bounded real...'** We treat $x_1^N$ as a finite list of non-random real numbers, so the only source of randomness is in the WoR sampling distribution (Section 1.2). Any finite list of numbers $x_1^N$ is bounded by definition, and all past work on WoR sampling assumes that these bounds are known (e.g. [13, 14, 15] in the paper's references). Relaxing this assumption is an interesting question for future work.

**(R4) 'There lacks several right bracket...'** Thank you, we have now fixed this.

**(R4) 'The notation J in Thm 3.1 appears abruptly'** We have now tried to make the role of $J$ less abrupt.

[Meta-Review · NeurIPS 2020]

This paper develops confidence sequences in the setting of sampling without replacement. I want to highlight some particular strengths of the paper. (*) The reviewers are in agreement that the writing is excellent. (*) The work is theoretically strong and practically relevant. (And yet the authors are also very clear to the reader what they are accomplishing in each bit; the prose is easy to follow.) (*) The theory approach is novel and interesting in its own right. (*) The authors start with great motivating examples (election opinion surveys, permutation tests, Shapley values, understanding the effect of interventions).